# Exploitation of Pigs during the Late Medieval and Early Modern Period in Croatia

Kim Korpes [1],*, Aneta Piplica [2], Martina Đuras [1], Tajana Trbojević Vukičević [1] and Magdalena Kolenc [1]

1   Department of Anatomy, Histology and Embryology, Faculty of Veterinary Medicine, University of Zagreb, Heinzelova 55, 10000 Zagreb, Croatia; martina.duras@vef.unizg.hr (M.Đ.); tajana@vef.unizg.hr (T.T.V.); mkolenc@vef.unizg.hr (M.K.)
2   Department of Animal Breeding and Livestock Production, Faculty of Veterinary Medicine, University of Zagreb, Heinzelova 55, 10000 Zagreb, Croatia; apiplica@vef.unizg.hr
*   Correspondence: kkorpes@vef.unizg.hr

**Abstract:** This study investigated the historical consumption of pork in Croatia through a comprehensive analysis of pig bones from six medieval archaeological sites, comprising three castles and three monasteries dating from the 13th to the 16th century A.D. Employing a detailed morphological analysis of pig bones and teeth, the study quantified the number of identified specimens (NISP) per site. On bones and teeth, identification of sex and age was performed, and butchery patterns were documented. The results revealed a significant presence of pork in the diet of castle and monastery inhabitants, with pig bones being the most abundant animal remains. Age analysis suggested a prevalence of juvenile pigs, predominantly males. The findings implied that from the 13th to the 16th century A.D., pork was a staple in the diet of castles and monasteries in continental Croatia, sourced from pig breeding outside these sites. Typically, male pigs aged under two years were delivered for consumption to these establishments.

**Keywords:** exploitation; pig; medieval; modern; archaeozoological material; Croatia





## 1. Introduction

Pigs were used as the source of meat and fat in inhabitants' diets throughout ancient Europe. There are many controversies about the use of pork during the Medieval Period on the European territory due to different influences of social, religious and economic factors. In the Late Medieval Period, inhabitants of that area lived in high-status sites, i.e., fortresses and castles; low-status sites, i.e., rural areas; and religious sites, i.e., monasteries. Urban sites were inhabited by diverse populations originating from high and low statuses. Everyday life differed remarkably among the sites, including significant differences in diet and food production. Throughout medieval Europe, animal production was located mainly in rural areas [1,2]; in urban sites, animal production was confirmed, but to a much lower extent. It is presumed that the animal production in urban sites was not enough to satisfy the needs of their inhabitants [3], and, therefore, animals were imported from rural areas. Contrary to rural and urban sites, high-status sites were determined as exclusive consumption sites. Consequently, the trade became a significant social act where, together with other animal products such as wool, milk and fat, meat was transported from the site of production to the site of consumption.

By analysing animal remains from a certain archaeological site, it is possible to distinguish sites with developed animal production from exclusive food consumption sites. For an accurate analysis, animal remains need to be classified according to the sex and age at death. Food consumption sites show a higher abundance of remains originating from male and younger animals [4,5]. A high presence of long bones from the proximal parts of the pelvic and thoracic limbs indicates exclusive consumption sites, since these parts represent the fleshiest part of the animal body. On the other hand, animal production sites

include remains originating from male and female animals of an older age; however, a small number of young animals may be found. The latter should also include bones from neonatal individuals, indicating stillbirths that appear normally during the production process [6]. Animal production and consumption has been analysed based on animal remains at numerous archaeological sites throughout Europe [7–10].

The main reason for pig domestication was meat and, to a lower extent, fat production, which is also the reason for recent pig breeding. Namely, pigs are animals that, except for meat and fat, provide very few secondary products. Compared to ruminants, which were used for milk and meat production and as working animals, the age at death of ancient pigs was usually lower [11]. Pigs were slaughtered between their first or second and third year of life when they reached the optimal weight [4]. Such young animals are represented by juvenile, unfused bones in the archaeozoological material, which makes the interpretation of pig remains, in terms of their exploitation, consumption and breeding, sometimes difficult. Moreover, distinguishing between wild and domestic pigs is particularly challenging compared to other species [12].

The territory of Croatia historically represents a crossing of important European trade ways, religions and cultures. Out of the various types of animal production, our study focuses on pig meat production and pork consumption during the Medieval and Early Modern Period on the territory of Croatia in three monasteries and three high-status sites. In Croatia, a comprehensive analysis of pig bones from archaeological sites has not been performed yet; however, small-scaled research on the diet of ancient inhabitants based on animal remains has been performed [13–16]. To date, comprehensive insight into the pig breeding, production and pork consumption in rural and urban areas and high-status sites of that region is still missing, to the best of the author's knowledge. Unfortunately, there is a lack of medieval cookbooks and very few preserved cost lists of the nobility and clergy from the Croatian territory.

Our study aims to determine the abundance of pig bones in the animal material originating from six medieval archaeological sites in Croatia. We will consider the importance of pork in the diet of the medieval nobility and clergy in Croatia, as well as explore the ways in which they utilized the carcasses of these animals.

## 2. Materials and Methods

Excavations at six archaeological sites, performed by the Croatian Conservation Institute and Institute of Archaeology, both situated in Zagreb, Croatia, revealed extensive archaeozoological material. The excavation of animal remains took place in 2010, 2014, 2018 and 2019, and it is estimated that the material originated from the Late Medieval Period (13th century A.D. and beginning of the 14th century A.D.) and the Early Modern Period (15th and 16th century A.D.). Archaeological sites included in this study range geographically from West, over to the North and then to the East continental part of Croatia (Figure 1). The archaeological site Plemićki grad Vrbovec (abbrev. PGV) in Zagreb County includes remains of a Romanesque burg with a perimeter wall that was inhabited from the 13–15th century A.D. [17]. Stari grad Milengrad (MIL) is located in Krapina-Zagorje County and was built in the 14th century A.D. and served as a medieval manor [18]. The Pauline monastery of All Saints in Streza (STR) is in Bjelovar-Bilogora County and represents remains of the monastery built in 14th century A.D. [19,20]. The Benedictine Monastery of St. Margaret in Bijela (BSM) was one of Bjelovar-Bilogora County's important benedictine centres in the 14th and 15th centuries A.D. Today, it is represented by remains of the monastery on an elevated ground above a stream surrounded by a defensive wall [21]. The most eastern site is the Benedictine monastery of St. Michael in Rudina (RUD), located in Požega-Slavonia County and represented by remains of the abbey from the 13th century A.D. [22,23]. Located on a peninsula in National Park Plitvice Lakes is Stari grad Krčingrad (KRC), consisting of a triangular tower surrounded by a defensive wall. It was built from the end of 13th and the beginning of the 14th century A.D. [24]. BSM, RUD and STR were monasteries of Benedictine and Pauline monks, while KRC, MIL and PGV were castles.

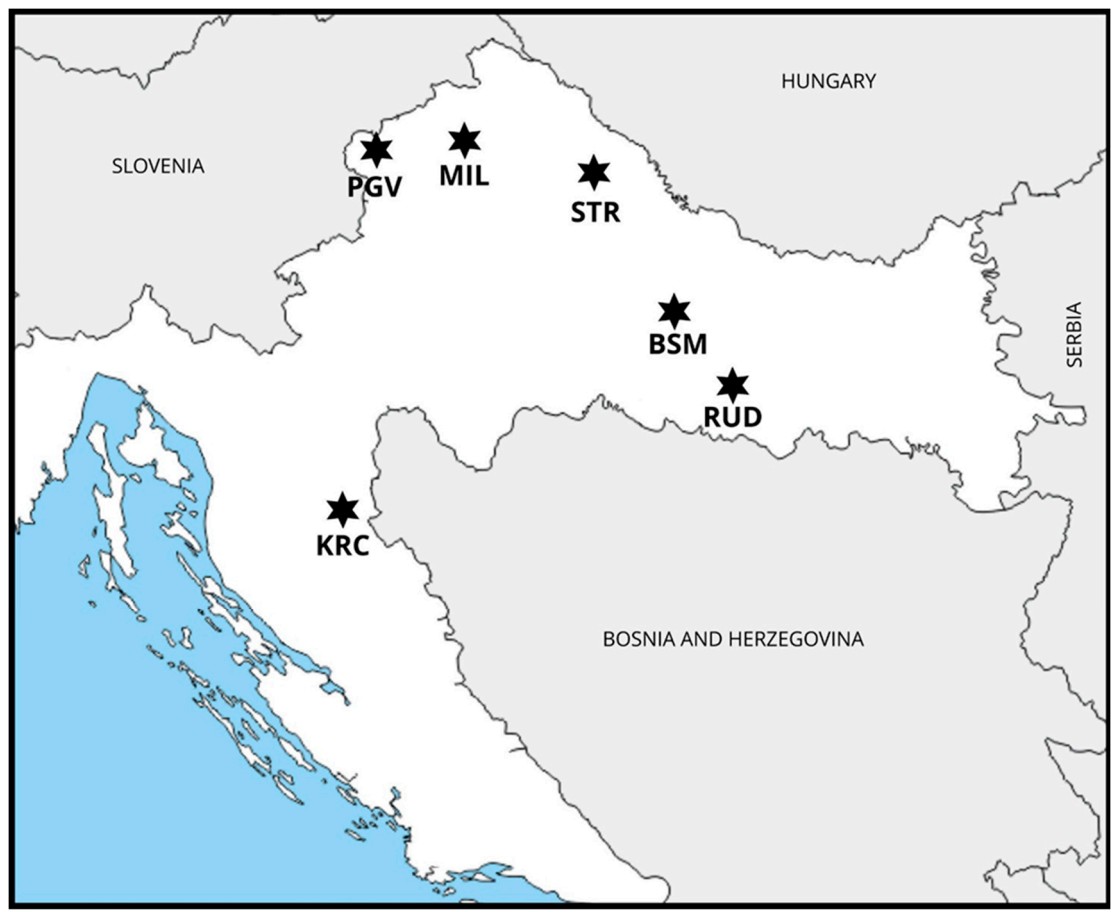

**Figure 1.** Map of Continental Croatia shows locations of investigated archaeological sites (asterisk). BSM—Benedictine monastery of St. Margaret, KRC—Stari grad Krčingrad, MIL—Star grad Milengrad, PGV—Plemićki grad Vrbovec, RUD—Benedictine monastery of St. Michael, STR—Pauline monastery of All Saints.

Excavated archaeozoological material stored in plastic bags was delivered to the Archaeozoological Laboratory in the Department of Anatomy, Histology and Embryology at the Faculty of Veterinary Medicine, University of Zagreb, Croatia. Specimens for each archaeological site, after washing and drying, were divided into identifiable and unidentifiable groups. Identifiable animal remains were categorized following Miracle and Pugsley's criteria [25] (p. 260). These categories included all teeth and their fragments, fragments exceeding 5 cm in length, fragments with portions of the articular surface, diaphyses of long bones displaying nutrient foramina and/or noticeable muscle attachments, as well as fragments of the skulls and mandibles with identifiable bone morphology. All identifiable remains were subjected to skeletal and taxonomic identification using the veterinary osteology and archaeozoological literature [26–28], as well as comparison with a collection of recent bones from domestic and wild mammals. Pig bones and teeth were isolated from the archaeozoological material and subjected to further detailed morphological analysis.

To estimate the number of animal specimens represented at each archaeological site, the number of identified specimens (NISP) per site was quantified [29] (p. 27). The frequency of each animal species was expressed by absolute number and percentage of NISP, for example NISPpig from the absolute number of pig remains and %NISPpig for the frequency of those remains. Bones were classified according to their skeletal position, thereby offering a comprehensive depiction of the distribution of skeletal elements across the investigated sites. The type of pig exploitation at a certain site was presumed based on the age of animals at death and the sex of specimens. Bones and teeth were included in age estimation. Mandibulae and maxillae with teeth were used for age estimation based

on the sequence of tooth eruption, replacement and tooth wear phases [27,30]. Mandibles featuring unworn second permanent molars were classified as juvenile, while those with the second permanent molars in wear and the third permanent molars not yet in wear were categorized as subadults. Mandibles where the third permanent molars were in wear were designated as adults. Age estimation based on bones was performed according to the fusion of epiphyses to the shaft [31] (p. 76). The bones were grouped into three categories: juvenile (where early fusing bones were unfused), subadults (middle fusing bones with visible epiphyseal line) and adults (where late fusing bones were fused). The sex of the animal was estimated based on the sexual dimorphism expressed on the canine teeth [27]. If present, traces of butchering, such as cuts and incisions, along the bone were recorded.

To further explore potential relationships within the data, Pearson's chi-square test was used to enable assessment of the statistical significance of associations between the number of determined pig bones and age and sex categories among the studied sites. Distinctions between monasteries (BSM, RUD, and STR) and fortified sites (KRC, MIL and PGV) were subjected to analysis. The criterion for statistical significance was established at a significance level of $p < 0.05$. All statistical analyses were performed using the statistical software TIBCO Software Inc. 14.1.0.8 [32].

### 3. Results

The analysis of the archaeozoological material from all six sites revealed an extensive amount of animal remains that consisted of bones, teeth, horns and antlers originating from domestic and wild animals. On most sites, domestic animal remains originating from pigs, cattle and small ruminants (goats and sheep) prevail over remains originating from horses, carnivores and poultry, wild mammals and birds (Figure 2). Pig remains were most numerous in the material from all studied archaeological sites except BSM. The archaeological site with the highest percentage of pig bones was RUD (43.7%), followed by STR (43.3%), whereas BSM had the lowest percentage of pig bones (12.1%). At this site, cattle and small ruminant remains prevailed over pigs (Figure 2).

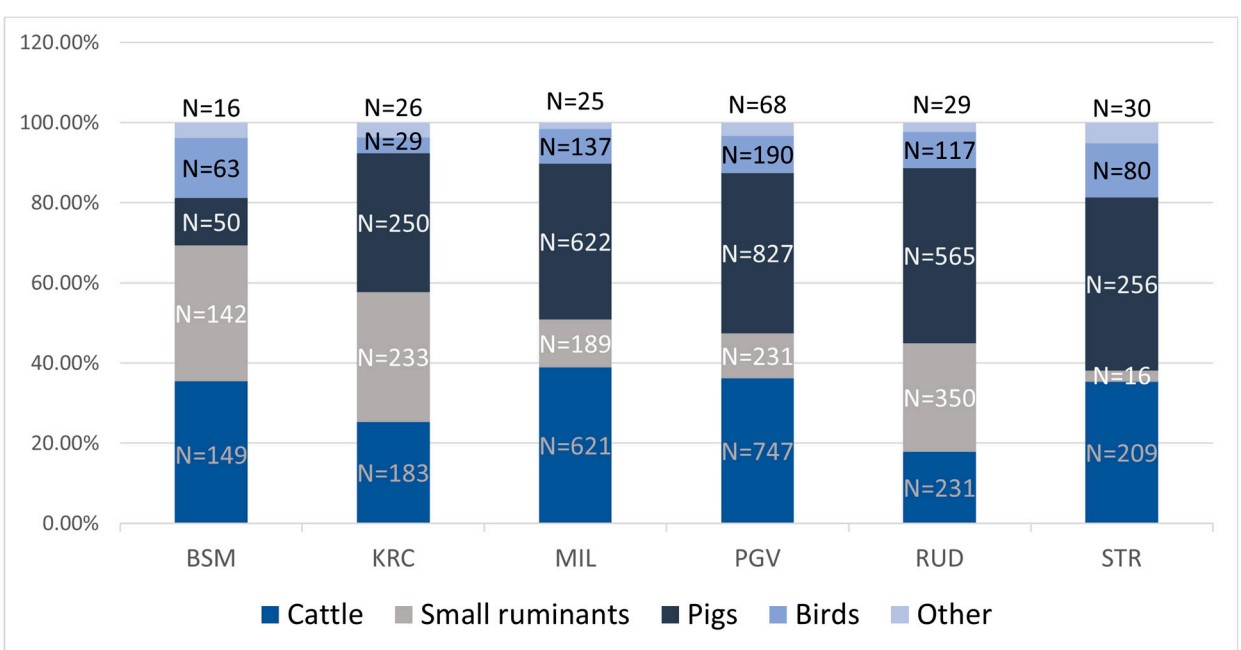

**Figure 2.** Abundance of wild and domestic animal remains expressed in numbers (NISP marked along the column) and as total sample ratio of the studied archaeological sites.

Wild mammals, carnivores, and horse, referred to as "Other", appeared in a small percentage at all the sites (Figure 2). Some of these animal species were represented by one

remain only in the whole sample, e.g., dog, mole and squirrel (Table 1). Such a low number of remains of a certain animal species was insufficient for further analysis of the abundance of these species at studied sites.

**Table 1.** NISP of wild mammals, carnivores and horses from six investigated archaeological sites.

| Archaeo Site/Species | Red Deer | Roe Deer | Hare | Squirrel | Vole | Horse | Bear | Cat | Dog | Mole | Ferret | Fox | Σ |
|---|---|---|---|---|---|---|---|---|---|---|---|---|---|
| BSM | 3 | 2 | 2 | 0 | 0 | 0 | 0 | 0 | 0 | 0 | 0 | 9 | 16 |
| KRC | 6 | 3 | 10 | 1 | 2 | 1 | 3 | 0 | 0 | 0 | 0 | 0 | 26 |
| MIL | 2 | 5 | 14 | 0 | 0 | 1 | 0 | 1 | 1 | 1 | 0 | 0 | 25 |
| PGV | 12 | 20 | 29 | 0 | 0 | 0 | 0 | 0 | 0 | 0 | 3 | 4 | 68 |
| RUD | 7 | 7 | 5 | 0 | 0 | 8 | 1 | 1 | 0 | 0 | 0 | 0 | 29 |
| STR | 8 | 6 | 15 | 0 | 1 | 0 | 0 | 0 | 0 | 0 | 0 | 0 | 30 |

Studied archaeological sites differed in the quantity of skeletal remains suitable for taxonomic and skeletal identification. The highest number of pig remains (NISPpig = 827, i.e., 39.8%) was determined at the archaeological site PGV, which was also the site with the highest number of determined skeletal elements (NISP = 2079). The lowest percentage of pig remains that showed statistical significance in the sample was at the BSM site (NISPpig = 414, i.e., 12.2%) (Table 2). Furthermore, statistically significant was the higher number of determined pig remains from the site RUD and all other sites, except site STR. When expressed as a ratio, RUD (43.7%) and STR (43.3%) had the highest percentage of pig remains.

**Table 2.** Number of identified animal remains (NISP) with NISP pig and %NISP pig per archaeological site.

| Archaeological Site | NISP | NISP Pig | %NISP Pig |
|---|---|---|---|
| **BSM** | 414 | 50 | 12.2% |
| **KRC** | 722 | 250 | 34.6% |
| **MIL** | 1595 | 622 | 39% |
| **PGV** | 2079 | 827 | 39.8% |
| **RUD** | 1293 | 565 | 43.7% |
| **STR** | 591 | 256 | 43.3% |

Animal remains were grouped according to their skeletal position, e.g., cranial bones, thoracic limb bones and pelvic limb bones. Skeletal categories showed different abundance in the total sample and samples per site. Most pig remains analysed in our study belonged to the thoracic limb bones (N = 815, Table 3, Figure 3), with the humerus (N = 256) being the most commonly recovered bone of the thoracic limb, while the scapula (N = 168), ulna (N = 148) and radius (N = 142) were less represented. Bones of the autopodium of the thoracic limb were represented with three samples for carpal bones and 98 for metacarpal bones. Out of all the cranial bones (N = 696), the mandibula ($N_{BSM}$ = 5, $N_{KRC}$ = 12, $N_{MIL}$ = 82, $N_{PGV}$ = 67, $N_{RUD}$ = 86, $N_{STR}$ = 26) was the most represented. The second most represented cranial bone was the maxilla (N = 175). Other bones of the skull, such as the incisive, occipital, temporal and zygomatic bones, were less abundant (N < 20). The total number of pelvic limb bones was lower (N = 592) than the number of thoracic limb bones, with the tibia being the most numerous bone (N = 185), followed by hip bones (N = 136) and femurs (N = 127). Bones of the autopodium of the pelvic limb were represented with 52 tarsal bones and 78 metatarsal bones. Only a small number of fibulae were preserved and identified (N = 14). The vertebrae were less preserved compared to other bone groups, i.e., cranial bones, thoracic and pelvic limb bones. Among all the phalanges, the proximal one (N = 37) was the most abundant, followed by the middle (N = 13); only four distal

phalanges were found. The bones of autopodium, including the metapodia and phalanges that couldn't be identified, were the least represented in the assemblage (N = 30).

**Table 3.** Number of identified remains grouped into skeletal categories per archaeological site. MTC—metacarpal bone, MTT—metatarsal bone, PH. PROX.—proximal phalanx, PH. MED.—middle phalanx, PH. DIST.—distal phalanx, Metapodium—metacarpal and metatarsal bones that couldn't be skeletally identified due to fragmentation.

| | Archaeological Site | | | | | | |
|---|---|---|---|---|---|---|---|
| **Bone** | **BSM** | **KRC** | **MIL** | **PGV** | **RUD** | **STR** | **Total** |
| **Mandible** | 5 | 12 | 82 | 67 | 86 | 26 | 278 |
| **Maxilla** | 3 | 13 | 43 | 63 | 41 | 12 | 175 |
| **Other** | 7 | 24 | 39 | 107 | 54 | 12 | 243 |
| **ΣCranial Bones** | **15** | **49** | **164** | **237** | **181** | **50** | **696** |
| **Individual Teeth** | **5** | **18** | **104** | **54** | **68** | **47** | **296** |
| **Vertebrae** | **0** | **34** | **12** | **26** | **11** | **4** | **87** |
| **Scapula** | 3 | 13 | 37 | 70 | 27 | 18 | 168 |
| **Humerus** | 3 | 7 | 72 | 72 | 70 | 32 | 256 |
| **Radius** | 4 | 13 | 32 | 43 | 35 | 15 | 142 |
| **Ulna** | 5 | 15 | 35 | 38 | 36 | 19 | 148 |
| **Ossa Carpi** | 0 | 3 | 0 | 0 | 0 | 0 | 3 |
| **MTC** | 0 | 19 | 10 | 51 | 11 | 7 | 98 |
| **ΣThoracic Limb Bones** | **15** | **70** | **186** | **274** | **179** | **91** | **815** |
| **Hip Bone** | 5 | 9 | 21 | 57 | 27 | 17 | 136 |
| **Femur** | 4 | 11 + 1 * | 38 | 41 | 18 | 14 | 127 |
| **Tibia** | 3 | 11 | 42 | 61 | 48 | 20 | 185 |
| **Fibula** | 0 | 1 | 3 | 5 | 5 | 0 | 14 |
| **Ossa Tarsi** | 0 | 13 | 10 | 16 | 10 | 3 | 52 |
| **MTT** | 1 | 1 | 19 | 40 | 11 | 6 | 78 |
| **ΣPelvic Limb Bones** | **13** | **47** | **133** | **220** | **119** | **60** | **592** |
| **Ph. prox.** | **0** | **16** | **7** | **9** | **5** | **0** | **37** |
| **Ph. med.** | **0** | **2** | **7** | **3** | **1** | **0** | **13** |
| **Ph. dist.** | **0** | **3** | **1** | **0** | **0** | **0** | **4** |
| **Metapodium** | **2** | **11** | **8** | **4** | **1** | **4** | **30** |
| **NISP** | 50 | 250 | 622 | 827 | 565 | 256 | 2570 |

* patella.

Out of the total number of pig remains, only 219 (8.5%) bones and 129 lower and upper jaws with teeth were suitable for age estimation. In the total sample, the juvenile group was the most numerous (N = 156). The highest number of juvenile pig bones was found at RUD (N = 48) and was similar to MIL (N = 41) and PGV (N = 36). The lowest number of bones belonging to juvenile animals (N = 3) was found at BSM, where the overall number of pig remains was the lowest (Table 4). Regarding bones from subadult animals, the total number of bones belonging to subadults from all the archaeological sites was 20, with the highest number found at PGV (N = 7), whereas, at BSM, no subadult pig remains were identified. The total number of bones from adult pigs was 43, with the highest number at KRC (N = 19), followed by PGV (N = 11). For the other archaeological sites, the number of adult pig bones was below 10 (Table 4).

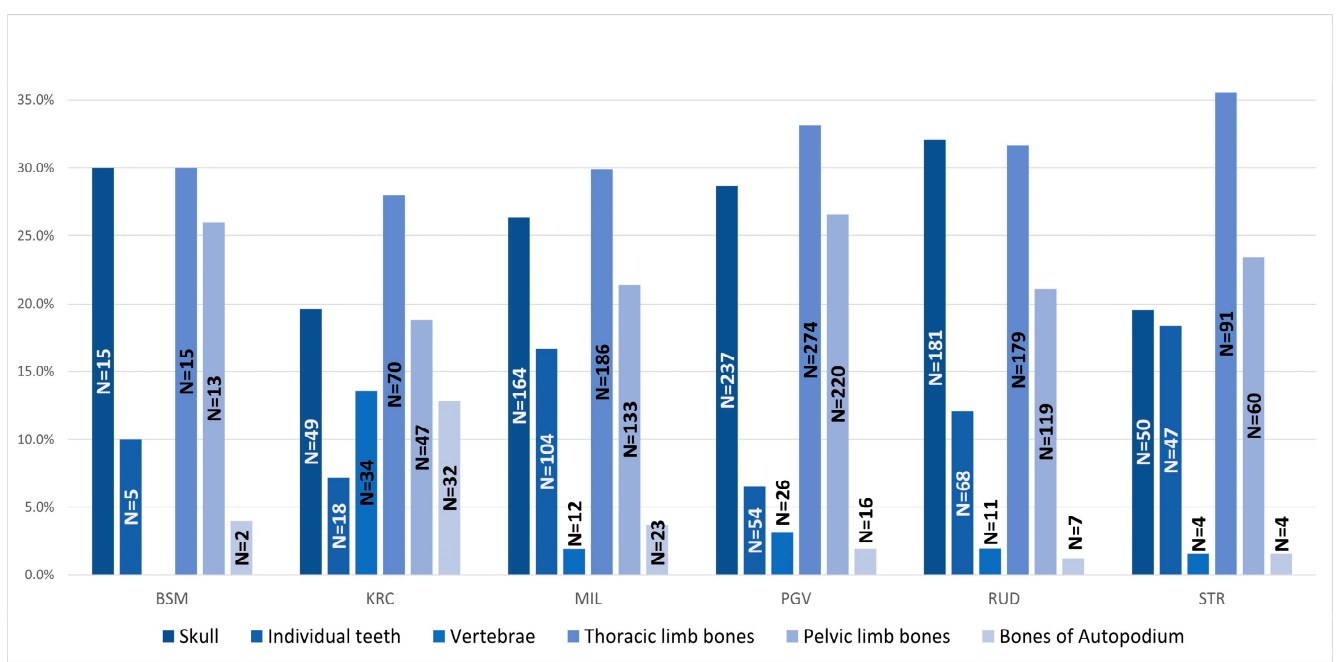

**Figure 3.** Frequency of skeletal groups identified at each archaeological site.

**Table 4.** Number of pig bones categorized into age groups.

| Arch. Site/Age Group | Juvenile | Subadults | Adults | Total |
|---|---|---|---|---|
| BSM | 3 | 0 | 2 | 5 |
| KRC | 19 | 2 | 19 | 40 |
| MIL | 41 | 5 | 4 | 50 |
| PGV | 36 | 7 | 11 | 54 |
| RUD | 48 | 4 | 4 | 56 |
| STR | 9 | 2 | 3 | 14 |
| Total | 156 | 20 | 43 | 219 |

The highest number of mandibulae and maxillae (N = 38) suitable for age determination was found at RUD, while the smallest number (N = 2) was recovered from the BSM sample (Figure 4). Based on teeth eruption and wear, three age groups were determined: juvenile, subadults and adults. At archaeological sites MIL, PGV and RUD, mandibulae/maxillae with teeth originating from all three age groups were found. At KRC, RUD and STR, most of the mandibulae/maxillae with teeth belonged to subadult pigs, which is the opposite of MIL and PGV, where most of the teeth belonged to adult pigs. At BSM, only two mandibulae with teeth originating from subadult pigs were determined and, therefore, they are not shown in Figure 4.

All the animal remains belonging to one of the age categories—juvenile, subadults and adults—according to epiphyseal fusion or teeth eruption and wear were summarized into the same age group in order to analyse the age of the pigs at death. The sites were also grouped into two categories according to their purpose, as determined by archaeological research (see Section 2): castles (KRC, MIL, PGV) and monasteries (BSM, RUD, STR). The number of juvenile and subadult pigs was significantly higher in monasteries than in castles when compared to the number of adults, i.e., the juvenile:adults ratio and the subadults:adults ratio was significantly different between castles and monasteries ($p < 0.05$). However, no statistically significant results were observed for the juvenile:subadults ratio.

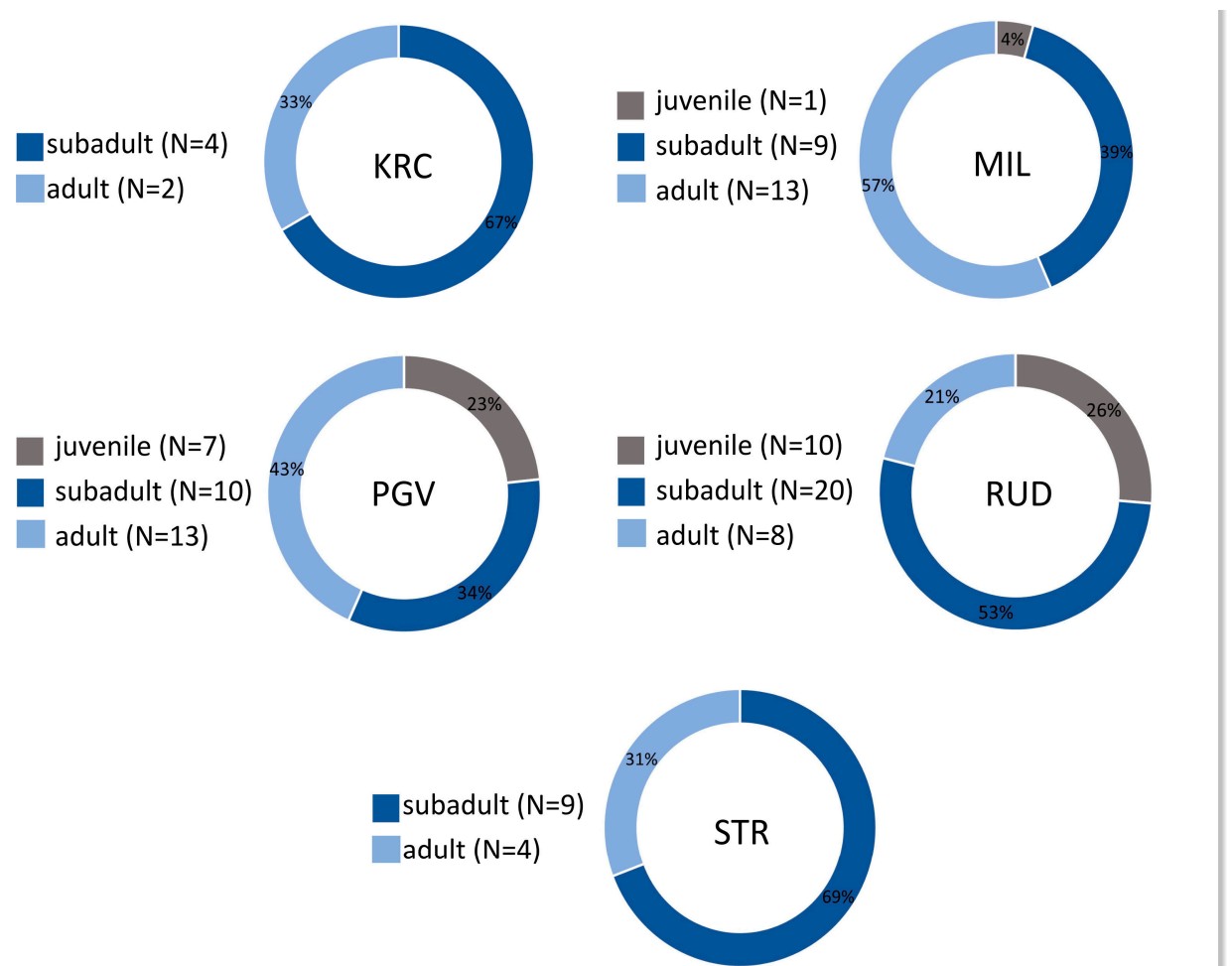

**Figure 4.** Ratio of age groups according to the teeth eruption and wear per archaeological site.

The sex could be determined exclusively by permanent canine teeth. In total, 82 mandibulae/maxillae with canines were recovered in the material from all the studied archaeological sites and were suitable for sex determination (Figure 5). Male canines (N = 67) were more numerous than female canines (N = 15) in the total sample, and male teeth prevailed over female on all archaeological sites, except on the site KRC, where two female and one male canine were recovered. On the site BSM, only male teeth were found (N = 3). The highest percentage of male teeth, in relation to female, was found on sites PGV (95.0%) and STR (71.4%), as shown in (Figure 5). The percentage of female canines remained under 40% on all sites with one exception: the percentage of female teeth on the site KRC (66.7%). When testing differences in sex ratio between castles and monasteries, the findings were not statistically significant. Sites BSM and KRC were excluded from the statistical analysis because of low or non-existent values.

Butchering marks were inspected on pig bones from the studied sample (Figure 6). The archaeological site KRC had the highest percentage, 28.8% (N = 72), of bones with butchering marks when compared to the number of identified bones in the sample from a site (NISP$_{KRCpig}$ = 250, 34.6%). The vertebrae were the most frequent (>64.7%) pig bones with butchering marks from KRC, whereas 48.5% were found on the pelvic limb bones. For all the other studied sites, the percentage of bones with butchering marks was less than 10%. At archaeological site STR, the percentage of bones with butchering marks was 9.8% (N = 25), while at MIL it was 8.8% (N = 55). For the archaeological sites BSM, PGV and RUD, the percentage of bones with butchering marks was 8% (N$_{BSM}$ = 4, N$_{PGV}$ = 66, N$_{RUD}$ = 45). At the archaeological sites MIL, PGV and RUD, most of the butchering marks

were discovered on the vertebrae, with a smaller proportion found on the bones of the thoracic and pelvic limbs. This contrasts with the findings at BSM and STR, where no butchering marks were observed on the vertebrae; instead, they were predominantly found on the bones of the thoracic and pelvic limbs.

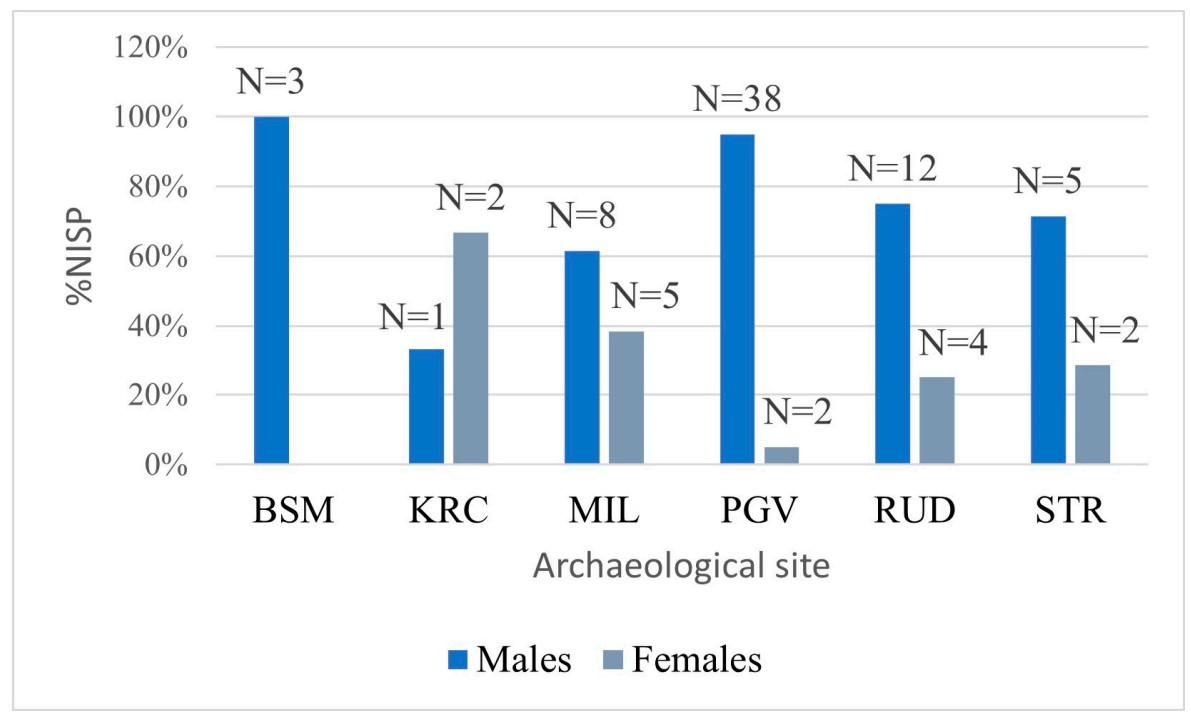

**Figure 5.** Ratio of male and female canines per archaeological site (N for each site and sex is showed on the outside end of the bar).

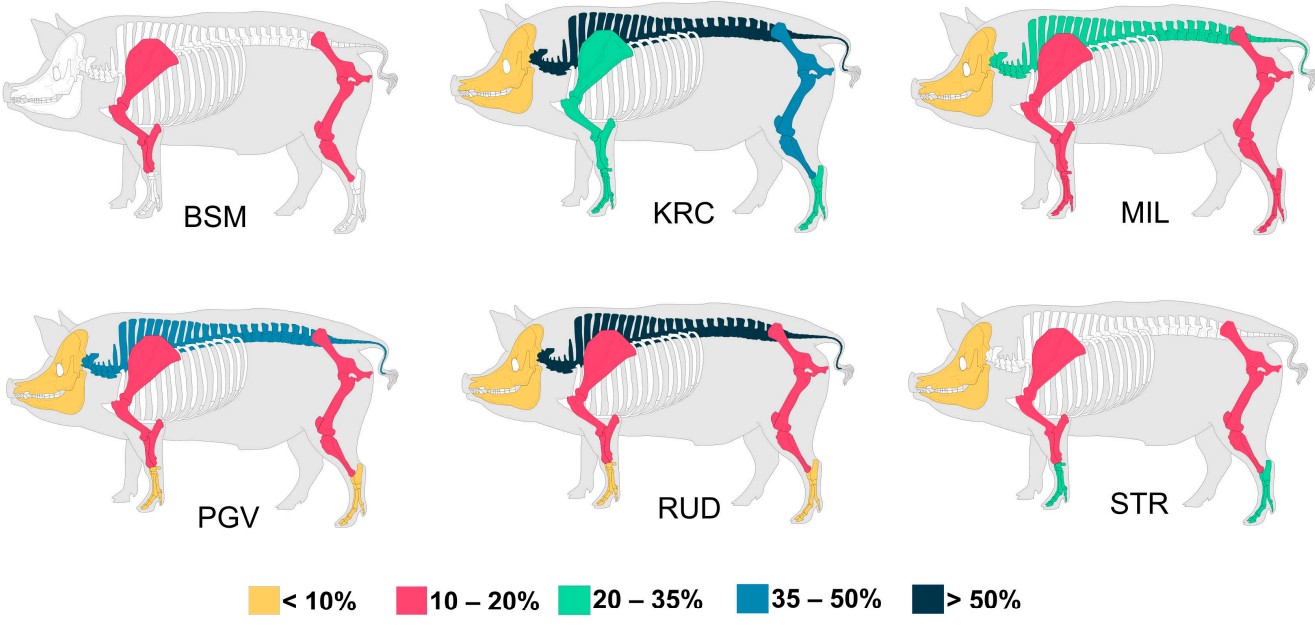

**Figure 6.** The distribution of butchering marks on pig bones.

## 4. Discussion

The social and economic status of a certain society is mirrored through the diet, which makes archaeozoological material the crucial evidence for sociological arrangements that were in force at a certain archaeological site.

### 4.1. Diet in Medieval Croatia

Our study indicates that pork was highly represented in the diet of inhabitants of monasteries and castles settled in the Medieval Period on the Croatian territory, which was revealed by the fact that pig bones represent the most numerous animal remains at all studied archaeological sites, except BSM. Our results are in concordance with the research on animal remains from archaeological sites on the Croatian territory done by Tkalčec and Trbojević Vukičević [16]. The research was focused on four high-status sites in medieval Slavonia, a Croatian region, and authors concluded that pigs seemed to be an important species in the diet of nobility. At the site Grubišno Polje-Šuma Obrovi, the pig was the dominant species (55.4%). At Sveta Ana-Gradina, two species dominated—pig (36.4%) and chicken (34.2%). On the site Veliki Zdenci-Crni Lug, the most represented species was cattle, followed by pig. Likewise, at Osijek Vojakovački-Mihalj, cattle was the most represented species (40.5%), followed by pig (19.1%). The authors pointed out the site Sveta Ana-Gradina and characterized it as a high-status site where young pig and chicken dominated the diet of the inhabitants [16]. Other research done on 1 723 mammal remains from the fortress Paka in northwestern Croatia revealed that pigs' meat dominated the diet of inhabitants (NISP = 1 024, 59.4%). As in our research, the results of analysis by Radović [15] did not determine pig breeding inside the fortress but presumed that whole pig carcasses of young animals were brought to the fortress. However, at some previously studied sites in Croatia, pig was not the dominant species. The diet of medieval inhabitants of five settlements (6–14th century A.D.) in Torčec-Gradić in Slavonia, Croatia, was based on pig and cattle meat [13]. Pig remains were the second most represented (38.8%) at the castle Čanjevo [14]. Research on the castle Barilović, included bones from three phases: the 15th and 16th centuries A.D. (feudal phase), the 17–19th century A.D. (military phase) and the end of the 19–20th century A.D. (civil administration). In all three studied phases, cattle were the dominant species and represented more than 50% of the archaeozoological material in all three phases, whereas pigs represented less than 20% [33].

Archaeological research and scarce written resources from Croatia state that, for castles, one of the sources of income was the tithe of pigs. However, there is no mention of pig breeding inside castles. It is known that monasteries were surrounded by forests with the aim of separating the clergy/monks from the world. This forest also served as a source of income because of pig farming. Farmers that lived near the Pauline Monastery of All Saints in Streza could use the forests belonging to the monastery for rearing pigs on acorns but had to pay for this privilege. Some farmers also resided in the monastery and took care of the pigs, along with other animal species, but no written evidence mentioned pig breeding [21,34].

### 4.2. Pork Consumption on European Sites

A study on animal material originating from 29 archaeological sites in Hungary, dated from the 9th to 17th century, showed opposite results. Beef and mutton prevailed in the diet of inhabitants, but, to the end of the Medieval Period, the consumption of pork increased, presumably because of the influence of German settlers, and, finally, pigs contributed to the diet at high-status sites more than mutton [7]. At Hungarian sites with Islamic influence, no pig bones were excavated at all [7]. In Italy, animal remains from the fortified site Vaccarizza, dated from the 10th to 13th century A.D., also revealed that pigs were the most represented species (37.1% until the end of the 11th century A.D.; 48.6% until the end of the 13th century A.D.). Other species were also consumed at this site, such as chicken, goose and sheep, but to a lesser extent [10]. Another study from Italy showed that different domestic species dominated in the diet of the inhabitants. Sheep and goats dominated

in the animal remains from Via Ginevra, pigs dominated at Palazzo Ducale and beef at Santa Maria delle Grazie, which were all urban sites. Pigs dominated also in the animal remains of the castle Castelluzzo di Molassana, followed by small ruminants and cattle [9]. Pig bones were the most abundant animal remains during the High Middle Ages in the Medieval castle Lanzekirchen, Lower Austria. However, cattle bones dominated in the material from the Late Middle Ages [35]. Based on the research of Bavarian castles from the Middle Ages, Pasda [36] observed that the pig bones of young animals were dominant in the material from sites with high status and/or great political importance.

### 4.3. The Influence of Saint Benedictine Rule?

The significantly low number of pig remains at BSM may be because this site was a Benedictine monastery, where inhabitants followed certain rules set by Saint Benedictine. Among those rules, a food rule says that it is forbidden to eat meat of quadrupeds except in the case of illness, whereas poultry meat is approved. The findings on site BSM is in collision with site RUD, which was also a Benedictine monastery. Here, we found the highest percentage of pig remains in the site sample, which could support our presumption that not all monks followed that food rule very strictly, or the rule was followed only on certain occasions. Studies from other countries show different results concerning the dietary habits in monasteries. For example, an analysis of two religious sites (archbishop's castle, 9–10th and 12–15th century A.D. and cloister, 13th century A.D.), three urban sites (residencies, 12–13th and 13–14th century A.D.) and one castle (16th century A.D.) revealed opposite results for the two religious sites excavated in Italy. In the archbishop's castle, a high-status diet was based on pigs, whereas the humble diet of monks in the cloister was based on poultry, sheep and goat meat [9]. Findings in the Dominican Monastery of Norden, Germany (15th till early 16th century A.D.) showed that their diet was based primarily on cattle, sheep, goat and pigs [37]. Another explanation for the low number of pig bones at BSM could be the Ottoman occupation that happened mid-15th century A.D. on that part of Croatian territory. Written sources state that the monastery was used by the Ottomans, who were Muslims [21] and practiced food rules of their religion, which exclude pork.

### 4.4. Pig Production Practices and Meat Consumption

Out of the total sample, only a small number of bones from our study was suitable for age determination. Age determination showed that most of the studied bones originated from juvenile pigs. Previous studies showed that in Medieval England, pig production was located mainly in rural sites, from which animals were transported to urban and high-status sites, where they were slaughtered, and the meat was eaten fresh [38]. The study shows no evidence of pig production and breeding inside castles or monasteries and there was only minimal evidence of pig production inside of towns. In the Medieval castle Lanzekirchen, Lower Austria, pig bones were the most abundant animal remains and during the High Middle Age; most of these bones originated from pigs aged one and two years. It is presumed that farmers brought only male pigs useless for breeding to the castle [35]. Excavations at St Gregory's Priory, Canterbury, UK (12–16th century A.D.), showed that pigs at the site were mostly slaughtered under two years of age and were predominantly males [39]. At Eynsham Abbey, UK (13–16th century A.D.), bones from juvenile and subadult pigs prevailed over adult ones, and both boars and sows were present. The authors presumed that inhabitants of the Abbey produced its own bacon [40]. Additionally, an indication of high-status diet is a high number of remains originating from animals that were killed before their optimal age for slaughter. However, the presence of young animal remains in the archaeozoological material is not exclusively a sign of high-status diet. Older animals can also be found in surplus on a site because of their production of fat, which was considered a high-status food [41]. Our study showed a high number of juvenile pig bones at both site categories, i.e., castles and monasteries, but no statistical significance was found between them. Therefore, we presume that the

high absolute number of bones originating from juvenile pigs indicate the consumption of juvenile pigs at the studied sites, whereas the low number of adult pig bones may indicate that pigs were not bred as a primary activity at the studied sites. Male canine teeth, which were used for sex determination, prevailed over female canines on all the studied archaeological sites, which also indicates consumption over production at the studied sites. We can presume that pig breeding probably occurred at rural sites near the castles and monasteries studied in this research. In our sample, the most abundant were the bones of proximal parts of the limbs and bones of the head. Proximal limbs are the fleshiest part of the pig body, showing the consumption of high-quality meat at certain sites. This is in concordance with the findings of Kühtreiber [35], who determined numerous proximal bones of bovine limbs at the Late Middle Age castle Lanzekirchen, Austria, and presumed that this is due to the high amount of meat on those body regions. On archaeological sites in Italy, the most abundant bones of the pig were cranial bones, including the mandible [9]. Research of Esztergom castle revealed, in a sample of 12.3% pig bones, a high number of head bones, which could be because of the preference for this part of the animal [42,43]. Although most parts of the pigs' carcasses were present at St Gregory's Priory and Eynsham Abbey, the frequency of the head bones was relatively high, indicating that heads could have been bought separately for some occasions [39,40]. A high number of pig cranial bones indicates picking the parts of the animal according to one's own preferences, which also indicates a high-status diet at a certain site and was the case in our research. It is not surprising that we found a preference for head, because different animal products made from head meat and meat jelly such as head cheese is still commonly eaten in Europe. Otherwise, in low-status sites, i.e., rural household a whole animal was used [41,42,44].

Butchering marks were found on bones originating from all the studied sites. Although vertebrae were less abundant than bones of the limbs and head, they showed a remarkable number of butchering marks compared to other bones. Nevertheless, butchering marks were also present on limb bones. Cut marks on vertebrae usually represent primary butchery, the result of which is the animal body being divided into smaller parts that are easier for manipulation and/or transport. Disarticulation is quite difficult to perform, especially between vertebrae, which could explain the highest number of cut marks being found on the vertebrae. We presume that young pigs, with lower weight, were brought to the sites as whole animals, while older pigs, i.e., subadults and adults, could have been primarily butchered on the site of production. Primary butchery, in this case, would include cutting the animal from the head to the tail along the median line, creating two halves of the carcass. Secondary butchery or final butchery was performed at the consumption site, which means that the cut marks, found on long bones of limbs, such as the humeri and femora, were there because of disarticulation or cutting into smaller parts suitable for further processing, removing meat or filleting the animal [45].

## 5. Conclusions

Pigs were the most important source of meat for inhabitants of the three monasteries and three castles from the Late Medieval and Early Modern Period studied in this research. The only exception was the archaeological site BMS, where the low number of pigs could have several explanations based on food rules implemented by the inhabitants. We did not find solid evidence of pig breeding inside the castles and monasteries included in our research. Therefore, we presume that the breeding took place in other places, outside castles and monasteries, while male pigs aged under two years were delivered to castles and monasteries as whole animals or halves for consumption.

**Author Contributions:** Conceptualization, K.K.; formal analysis, K.K. and A.P.; methodology, K.K. and M.K.; investigation, K.K. and M.K.; resources, M.Đ. and T.T.V.; writing—original draft preparation, K.K. and T.T.V.; writing—review and editing, A.P., M.K. and M.Đ.; visualization, K.K.; supervision, T.T.V. All authors have read and agreed to the published version of the manuscript.

**Funding:** This research received no external funding.

**Data Availability Statement:** All data used in the current study are available from the correspond-ing author upon reasonable request.

**Acknowledgments:** Our gratitude goes to the leaders of the excavations of archaeological sites included in this research who entrusted us with the archaeozoological material. Special thanks to Andrej Janeš (BSM) and Petar Sekulić (KRC, MIL, RUD, STR) from the Croatian Conservation Institute, Zagreb, and Tatjana Tkalčec (PGV) from the Institute of Archaeology, Zagreb.

**Conflicts of Interest:** The authors declare no conflicts of interest.

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
