# Peer review of "Exploitation of Pigs during the Late Medieval and Early Modern Period in Croatia"

_heritage, doi:10.3390/heritage7020049_

Round 1

Reviewer 1 Report

Comments and Suggestions for Authors

Dear authors, I was pleased to read such a concise study on the pig remnants in medieval Croatian sites. Such an initiative should be extended in the futureso it can cover the importance of the main species data for a geographical and historical era (where not so much data is available). On the other hand, as a veterinarian and (amateur?) archeozoologist I feel that the paper lacks one major set of data- the reconstructed morphology- the WRH data. I presume that this was not the main purpose of your approach, but, if the conditions allow this, it might be an important set of data that will make your approach complete and give others resources for complex comparisons (as you most probably would like to have in any archaeofaunal report). You should treat this not as a "must have" feature, but you should take it as a generic recommendation. Quality of graphs and tables is good, they seem to me quite sufficient and good. Regards and keep up the good work!

Reviewer 2 Report

Comments and Suggestions for Authors

This is a very important contribution to a better understanding of the importance of pigs and pork in the past in European societies. However, it is worth raising several methodological issues in the article that have not been published and reported sufficiently well. They are listed below.

Combining tooth ageing data with a fusion system is a problematic methodological approach because the latter method produces extremely inaccurate data. Therefore, providing detailed information in the supplement using both methods is recommended. Please describe how the age categories obtained from both methods were correlated.

The fact that the autopodium of the thoracic and pelvic limbs is considered together in the article is not an error, but the reason given for doing so is wrong. It is not true, written in lines 227-228, that "the thoracic or pelvic limb could not be determined due to the lack of morphological differences between the limbs”. Please compare (cf. Schmid 1972 - 1972 Atlas of animal bones for prehistorians, archaeologists and Quaternary geologists;  and also Driesch 1976 - 1976 A guide to the measurement of animal bones from archaeological sites.), As the routine, this is no problem for researchers working with subfossils animal remains recovered during excavation.

Charts are a graphical presentation of data contained in tables. The article includes both forms, which is unnecessary repetition. Please consider moving the tables to supplement the materials. In Figures as charts containing percentage data, the total size of each assemblage from which percentages are calculated should be provided. Please compare Figure 4 - this is well done. Do this in the following figures: 2, 3 (next to the site names), 5, 6.

Age at slaughter assessed based on teeth is an excellent tool for determining at least a dozen categories (e.g. Grant 1982: The use of tooth wear as a guide to the age of domestic ungulates; Benecke 1988 Archäozoologische Untersuchungen an Tierknochen aus der frühmittelalterlichen Siedlung von Menzlin. Materialhefte zur Ur- und Frühgeschichte Mecklenburg, Band 3. Schwerin). The two methods provided allow you to obtain detailed information that can be categorized as done in the article. However, it is necessary to explain the reason for the simplified categorization into juvenile, subadult and adult.

Please be careful when the sex ratio of pigs is assessed based on loose canines.

Canines of males, which are noticeably smaller than male tusks, are often overlooked during excavations, especially if the material is collected by hand. Therefore, it is recommended to examine the sex ratio based on specimens of mandibles and maxillae with preserved canines and/or canine sockets.

The use of acronyms for sites in the text and charts makes it very difficult to follow an analysis, particularly for readers from abroad. Please remove acronyms from the text, and consider doing this in figures.

In all figures, please increase the font sizes in captions and legends.

On the vertical axis (OY) in Figure 5, please provide a description of the data categories (n). 

Please do not enter percentages with an accuracy of 0.01. In zooarchaeology, enough is 0.1.

Reviewer 3 Report

Comments and Suggestions for Authors

Abstract: indicate the time periods as A.D.

Line 159: are small ruminants sheep and goats? If so, please say so.

Throughout results replace the uncommon word "detrition" with the common word "wear"

Line 298: insert "permanent" before canine teeth

The results should be written in present tense. This is important.

In Discussion section, add subheadings. The final section, regarding sites in Croatia, should come first.

I do not understand the final sentence of the Discussion section.

Comments on the Quality of English Language

Grammar:

Line 36, replace between with among

Throughout the paper, remove initial transition words like Additionally, Namely, Furthermore, On the other hand, Moreover, However ...

Lines 112 to 115 have a mix of singular and plural that needs correction. 

Line 263, change none to no

Reviewer 4 Report

Comments and Suggestions for Authors

The paper titled "Exploitation of Pigs during the Late Medieval and Early Modern Period in Croatia" is skillfully crafted, providing a thorough zooarchaeological analysis of the pig sample from the specified historical periods. The presentation and organization of the data are commendable. The authors demonstrate proficiency in interpreting the data, drawing comparisons not only with Croatian sites but also with contemporaneous archaeological sites in Germany, Hungary, Italy, Austria, and England. Furthermore, the authors delve into the taphonomy of the sample, comparing it to findings from other archaeological sites. In conclusion, this manuscript is well-executed and merits publication in its current form.

Round 2

Reviewer 2 Report

Comments and Suggestions for Authors

Dear Authors,

I accept the answers to the methodological and data presentation problems I have indicated. However, please note that age assessment based on fusion and teeth is still not shown separately. Such simplifications severely limit a more accurate understanding of age groups. According to numerous publications, pigs were slaughtered at least twice during the first year. How can one correlate the age given in closed time ranges (4-6 months; 6-10 months, 10-12 months) with the age that assesses the fusion of the epiphyses with the diaphysis (e.g. humerus 12-18 months fusion time, but if it is not fused, we have an open range, i.e. < 12-18 months, when is fused, we have again the open range > 12-18 months). In my opinion, such simplification goes too far. Unfortunately, data such as that contained in your work will not be helpful in comparative studies from other sites and periods when age assessment was performed and presented accurately. Zooarchaeology is excellent but only a guide - an introduction to the professional development of the methodology. On the other hand, considering your research's small age data set, I agree that the simplification method used in the article is sensible. Finally, I am convinced that the article is very important and interesting despite our differences in approach to methodological issues and data presentation. Congratulations.